# Nonsurgical Correction of Anterior Crossbite with Extra-Radicular Miniscrews—A Case Report

**Jae Hyun Park [1,2] and Johnny Joung-Lin Liaw [3,]***

1   Postgraduate Orthodontic Program, Arizona School of Dentistry & Oral Health, A.T. Still University, Mesa, AZ 85206, USA
2   Graduate School of Dentistry, Kyung Hee University, Seoul 130-071, Republic of Korea
3   National Taiwan University Hospital, Taipei 100225, Taiwan
*   Correspondence: ortho168@hotmail.com

**Abstract:** Protrusion can occur after correction of the anterior crossbite in Class III malocclusions. Four-premolar extractions might be indicated if the patients asked for a profile reduction. Two similar Class III anterior crossbite cases illustrate how the skeletal anchorage can prevent protrusion after anterior crossbite correction and the need for four- premolar extractions, as in the first case. The use of extra-radicular miniscrews at the infrazygomatic crest and buccal shelf are recommended for whole arch distalization of the maxillary and mandibular arches to reduce protrusion after anterior crossbite correction. It is an effective and efficient treatment alternative to extraction therapy for the protrusion after anterior crossbite correction. Long-term follow-up records show encouraging results supporting this paradigm shift in anterior crossbite correction with extra-radicular miniscrews.

**Keywords:** TSADs; anterior crossbite; Class III treatment; extra-radicular miniscrew

## 1. Introduction

Class III malocclusion is characterized by a composite of dentoskeletal patterns that lead to the forward positioning of the mandibular teeth in relation to the maxillary teeth and a concave profile [1–3]. In terms of etiologies, genetic factors are considered more influential than environmental factors [3,4]. The incidence of Class III malocclusions in the Asian population (14%) is much higher than that in Caucasians (1–5%) and the African American population (5–8%) [5–7]. The most common features of Class III malocclusion may include a retrognathic maxilla with proclined maxillary incisors and a prognathic mandible with retroclined mandibular incisors, although the components of Class III malocclusions may vary depending on the patient [1,2].

Therefore, treatment strategies for Class III malocclusion need to be adjusted depending on the age and severity of skeletal and dentoalveolar discrepancies. For adolescent Class III patients (CS1-3), a protraction facemask with or without RPE might be considered for orthopedic changes in the maxilla and zygoma [8,9]. Recently, bone- anchored maxillary protraction with a facemask or Class III elastics was suggested to minimize dental side effects in conventional maxillary protraction [10–14]. For adult Class III patients, if the discrepancy is mainly skeletal, orthognathic surgery might be the treatment of choice to improve facial profile with a stable occlusion [15,16]. When patients decline the surgical approach, and the degree of skeletal discrepancies is still within the range of camouflage orthodontic treatment; the orthodontist needs to determine whether the outcome will be protrusive after anterior crossbite correction [16]. If the answer is "yes", extraction would possibly be a better option for the treatment plan. With the help of TSADs, Class III malocclusion can be treated successfully using a nonextraction approach without the risk of subsequent perioral protrusion [17–22].

Temporary skeletal anchorage devices (TSADs) have been successfully applied in many clinical orthodontic cases, including maximal retraction in protrusion cases, Class

II correction, Class III correction, molar distalization in cases with crowding, molar intrusion in molar elongation cases, deep bite correction, open bite correction, midline correction, and the correction of canted occlusal planes and posterior crossbite [17,19,20,22–38]. Transverse correction with miniscrew-assisted RPE (MARPE), bone-anchored maxillary protraction, and active vertical control with TSADs have also been considered in the past decade [38–41]. BS miniscrews have been suggested to distalize mandibular dentition in Class III malocclusions [42–44]. This case report aims to demonstrate the need to use both IZC and BS miniscrews to reduce protrusion after anterior crossbite correction, which is a paradigm shift that greatly reduces the treatment duration and achieves a more pleasing facial profile after anterior crossbite correction [45–47].

## 2. Case 1

### 2.1. Diagnosis and Etiology

A 12-year-old female patient came to the clinic with her mother, complaining of maxillary anterior crowding and crossbites. There was no specific concern about her facial profile. The frontal facial photograph showed mild asymmetry, with her chin deviating to the right. The vertical proportions were harmonious and within the normal range. She had a straight profile with some lower lip eversion.

Anterior crossbites of all incisors and end-on Class III molar relationships were noted. She had severe crowding in the maxillary arch and mild crowding in the mandibular arch. Her overbite was 2 mm, while her overjet was −2 mm (Figure 1).

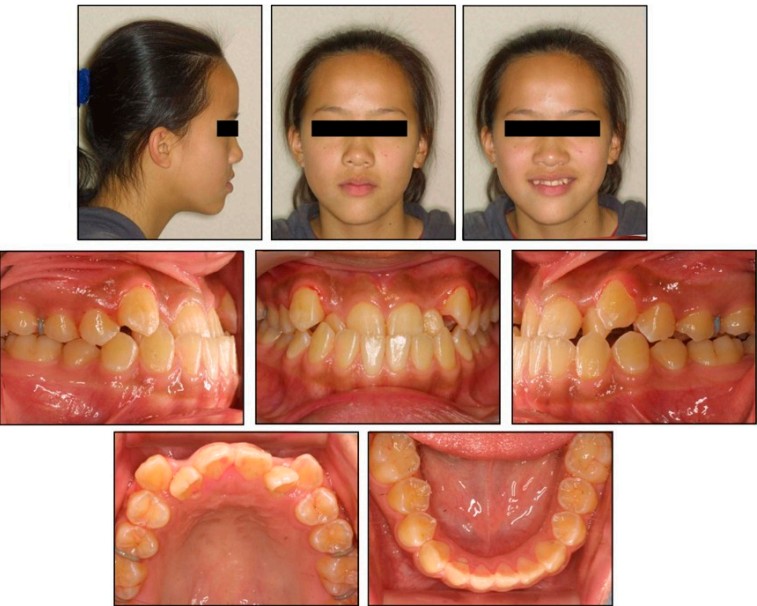

**Figure 1.** Case 1, Pretreatment facial and intraoral photographs.

A panoramic radiograph showed the presence of tooth germs for all her third molars. A lateral cephalogram revealed a Class III skeletal pattern (ANB, −2°; Wits, −11.8 mm). There were no severe dental compensations (U1-SN, 104.2°; L1-MP, 87.9°). The mandibular plane angle was within the normal range (MPA, 34.0°; FMA, 27.0°) (Figure 2, Table 1).

Her family history could not be traced to a Class III tendency; however, she was aware of an anterior crossbite in her primary dentition. The diagnosis of this patient was mild skeletal Class III with dental Class III relationships.

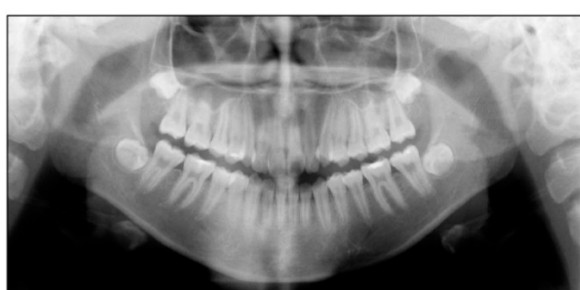
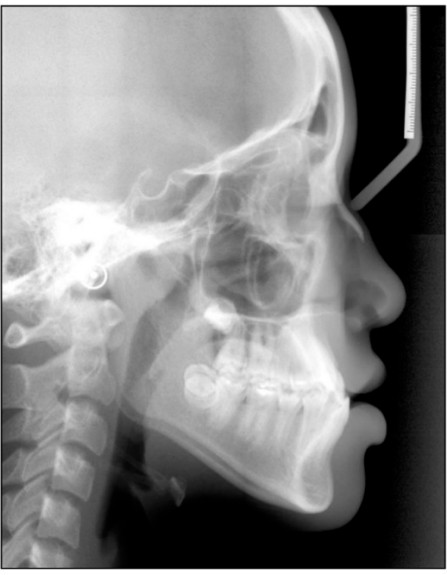

**Figure 2.** Case 1, Pretreatment panoramic radiograph and lateral cephalogram.

**Table 1.** Case 1, Cephalometrci measurements.

|  | Norms | Pretreatment | Posttreatment |
|---|---|---|---|
| Skeletal Analysis |  |  |  |
| SNA (°) | 81.5 ± 3.5 | 80.0 | 80.0 |
| SNB (°) | 77.7 ± 3.2 | 82.0 | 82.0 |
| ANB (°) | 4.0 ± 1.8 | −2.0 | −2.0 |
| SN-MP (°) | 33.0 ± 1.8 | 34.0 | 34.0 |
| Wits (mm) | −2.8 ± 3.3 | −11.8 | −8.9 |
| Dental Analysis |  |  |  |
| U1 TO NA (mm) | 3.9 ± 2.1 | 4.5 | 6.5 |
| U1 TO SN (°) | 108.2 ± 5.4 | 104.2 | 104.2 |
| L1 TO NB (mm) | 6.6 ± 2.8 | 5.0 | 2.5 |
| L1 TO MP (°) | 96.8 ± 6.4 | 87.9 | 74.6 |
| FACIAL ANALYSIS |  |  |  |
| E-LINE UL (mm) | −1.1 ± 2.2 | −1.0 | −0.5 |
| E-LINE LL (mm) | 0.5 ± 2.5 | 4.0 | 0.5 |

### 2.2. Treatment Objectives

Our treatment goals included correcting the anterior crossbites and Class III dental relationships while considering facial profile esthetics. As there was no significant arch length discrepancy in the mandibular arch and some residual growth might remain, we decided to correct the anterior crossbites first, then reevaluate the facial profile.

### 2.3. Treatment Plan

The treatment plan was to try a nonextraction approach first, with a reevaluation after the anterior crossbites were corrected. If the patient and her parents were satisfied with her facial profile, nonextraction treatment would continue to finalize the details. If they complained of lip protrusion during the reevaluation, we might have to shift to an alternative treatment plan with four first premolar extractions.

### 2.4. Treatment Progress

The treatment started with full mouth bonding of 0.022-in self- ligating brackets. (Damon 2, Ormco company, Orange, CA, USA) A bite turbo was bonded on the lingual surface of the mandibular right canine to prevent bracket interference and help anterior crossbite correction. A 0.014-in thermal nickel-titanium (NiTi) archwire was used as the

initial archwire on both arches. A few segments of NiTi open coil springs were inserted around the maxillary lateral incisors in the second month to create space for maxillary arch alignment. Both arches were well aligned after 4 months of treatment. Short Class III elastics were prescribed for Class III correction in the fifth month, with 0.016 × 0.025-in NiTi wires on both arches. We shifted to long Class III elastics in the seventh month with 0.019 × 0.025-in stainless steel wire on the maxillary arch. The occlusion looked almost like a solid Class I dental relationship after 8 months of treatment (Figure 3). As planned, we reevaluated the patient's facial profile after correcting the anterior crossbites.

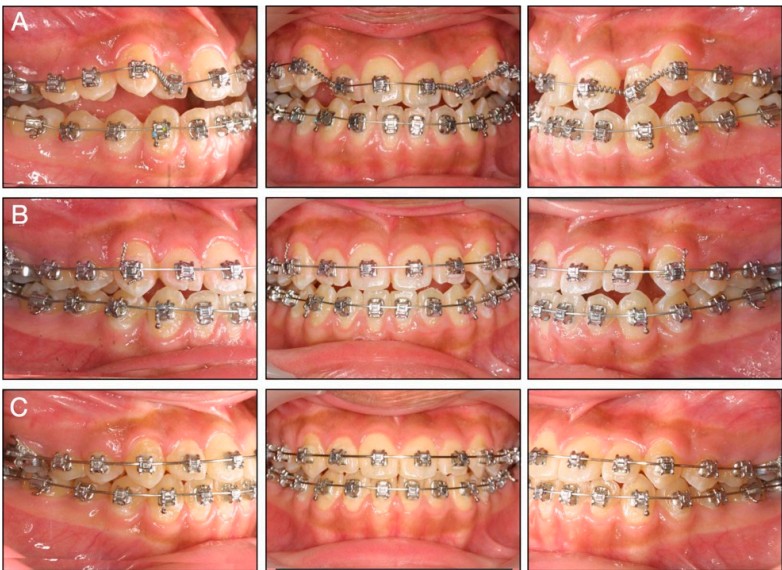

**Figure 3.** Case 1, Treatment progress at (**A**) 2nd month (**B**) 4th month (**C**) 8th month shows correction of anterior crossbite and Class III malocclusion.

A progressive record in the tenth month showed a borderline protrusive profile with very acceptable occlusion (Figure 4). After a thorough discussion with the patient and her parents, considering the facial profile, we decided to extract her four first premolars to reduce the protrusion. After extractions, space closure was accomplished on both arches with 0.019 × 0.025-in stainless steel wire. The fixed orthodontic appliances were removed after 25 months of active treatment (Figure 5). The patient and her parents were satisfied with her facial profile and occlusion.

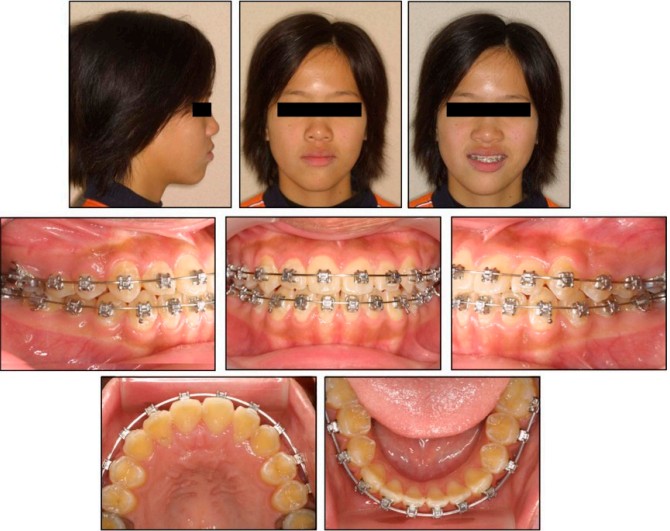

**Figure 4.** Case 1, Facial and intraoral photographs after 10 months of treatment.

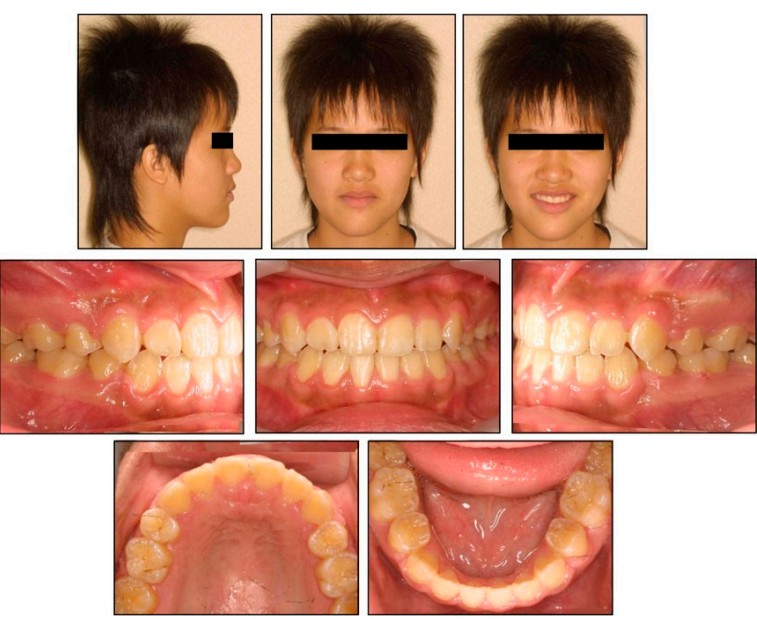

**Figure 5.** Case 1, Posttreatment facial and intraoral photographs.

*2.5. Treatment Results*

Both arches were well aligned with complete space closure. The occlusion fits well with good interdigitation and super Class I molar relationship. A posttreatment panoramic radiograph showed acceptable root parallelism without any apparent root resorption (Figure 6), but more work might be required on the mandibular second premolars to improve root angulation. A posttreatment lateral cephalogram showed a harmonious lip profile. Cephalometric superimpositions revealed only minor retraction of the mandibular incisors. Even with four premolar extractions, the maxillary incisors were moved slightly forward to maintain appropriate overjet with the mandibular incisors accompanied by mandibular growth (Figures 6 and 7). Progressive facial profile changes show an improved facial profile with the extraction approach (Figure 8).

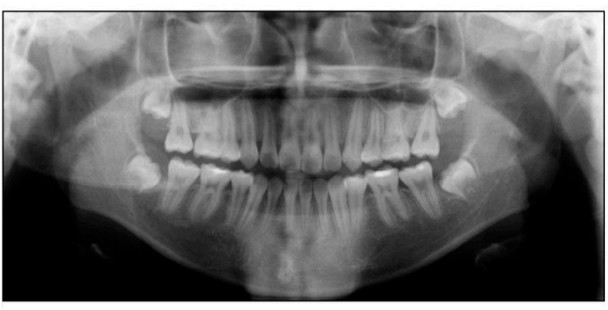
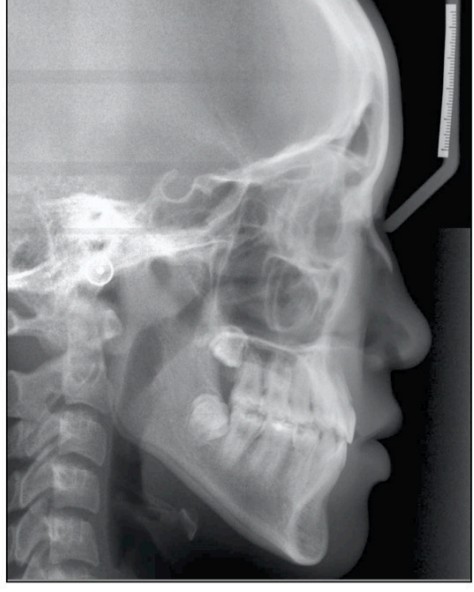

**Figure 6.** Case 1, Posttreatment panoramic radiograph and lateral cephalogram.

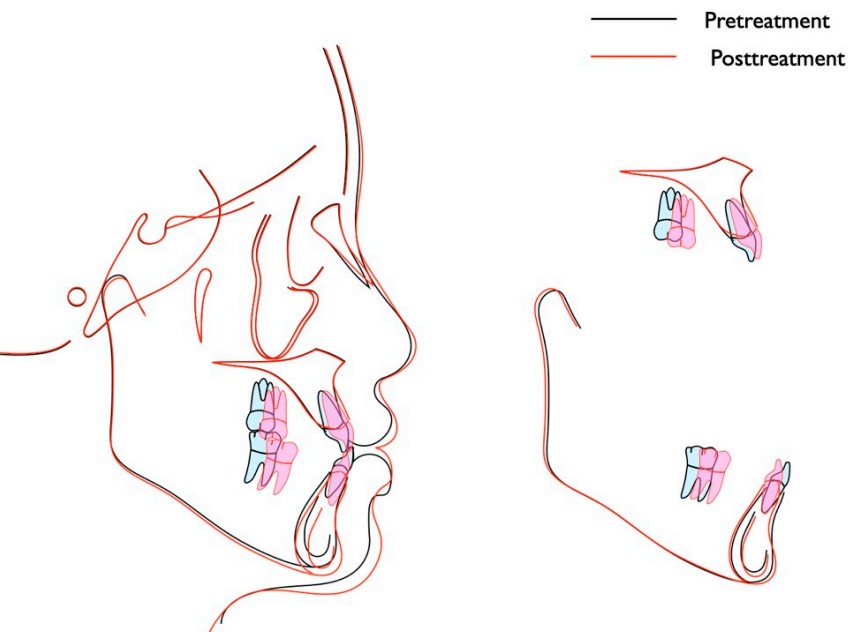

**Figure 7.** Case 1, Cephalometric superimpositions show the correction of anterior crossbite and Class III malocclusion with four premolar extraction approach.

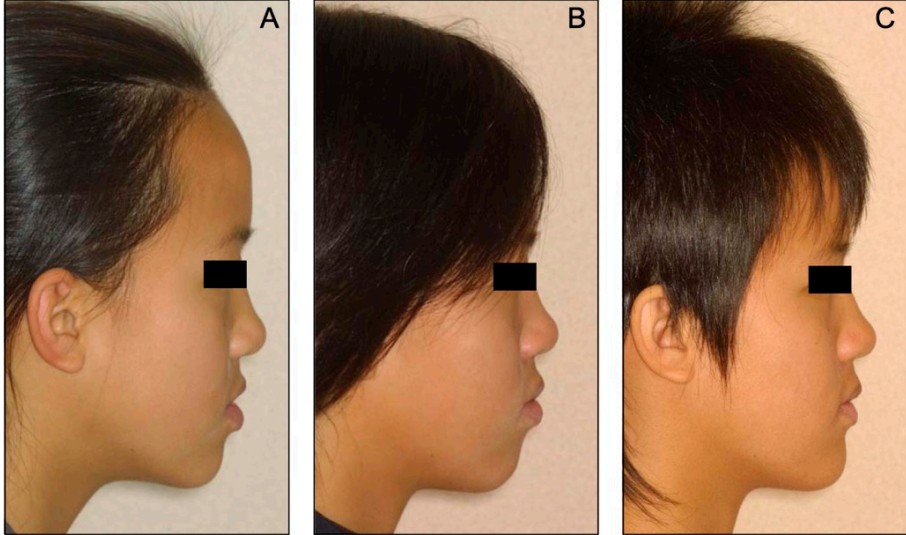

**Figure 8.** Case 1, Profile comparisons. (**A**) pretreatment (**B**) nonextraction approach (**C**) posttreatment with the four premolar extraction approach.

## 3. Case 2

### 3.1. Diagnosis and Etiology

A 24-year-old female patient requested orthodontic treatment with the chief complaint of mandibular prognathism and anterior crossbites. Her frontal facial photograph showed mild facial asymmetry with a slight chin deviation to the right. Her vertical proportions were within normal limits. Maxillary anterior malalignment was evident in her smiling facial photograph. The lateral profile view showed a concave profile because of mandibular prognathism (Figure 9).

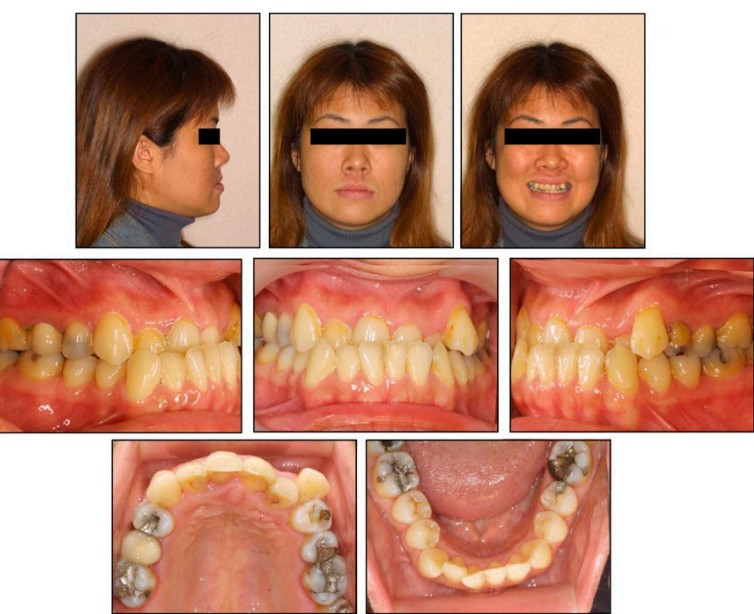

**Figure 9.** Case 2, Pretreatment facial and intraoral photographs.

Anterior crossbites and a deep overbite can be seen in the intraoral frontal photograph. There was no functional shift of the anterior crossbite to achieve an edge-to-edge relationship. Dental midline discrepancy was also noted. The maxillary dental midline was shifted to the right by 2 mm relative to the facial midline. The mandibular dental midline was further shifted to the right by another 2 mm relative to the maxillary dental midline. She showed severe crowding (approximately 6.5 mm) in the maxillary arch, and her maxillary right second molar was missing. The mandibular arch had mild crowding (approximately 1.5 mm). End-on, Class III dental relationships were noted.

A panoramic radiograph showed horizontal impaction of the mandibular third molars and evidence of endodontic treatment of the maxillary right central incisor and maxillary right second premolar (Figure 10). A lateral cephalogram revealed a skeletal Class III pattern (ANB, −4°; Wits, −12.7 mm). The mandibular plane angle was within the normal range (SN-MP, 28.5°). The maxillary incisors showed normal inclination, while the mandibular incisors were proclined (U1-SN, 105.7°; L1-MP, 82.0°) (Figure 10, Table 2).

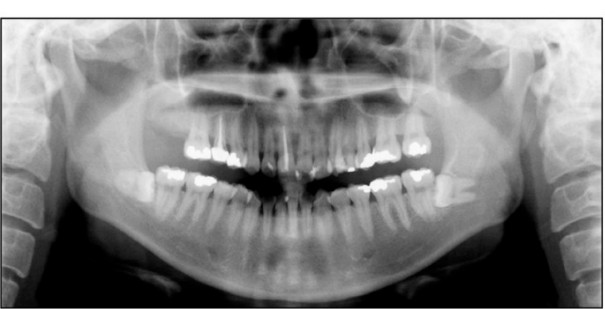
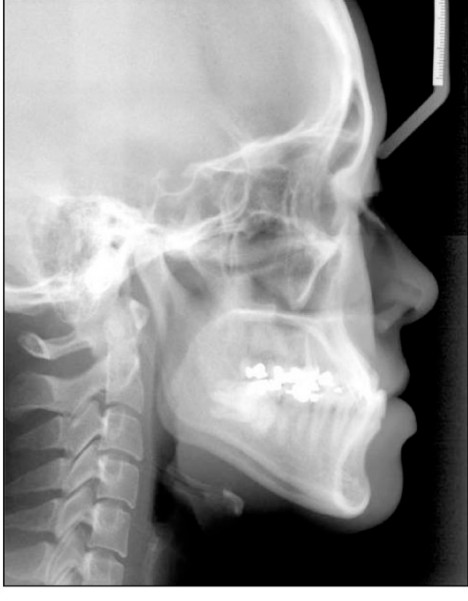

**Figure 10.** Case 2, Pretreatment panoramic radiograph and lateral cephalogram.

**Table 2.** Case 2, cephalometric measurements.

|  | Norms | Pretreatment | Posttreatment |
|---|---|---|---|
| Skeletal Analysis |  |  |  |
| SNA (°) | 81.5 ± 3.5 | 82.0 | 82.0 |
| SNB (°) | 77.7 ± 3.2 | 86.0 | 85.0 |
| ANB (°) | 4.0 ± 1.8 | −4.0 | −3.0 |
| SN-MP (°) | 33.0 ± 1.8 | 28.5 | 29.8 |
| Wits (mm) | −2.8 ± 3.3 | −12.7 | −11.4 |
| Dental Analysis |  |  |  |
| U1 TO NA (mm) | 3.9 ± 2.1 | 9.0 | 10.5 |
| U1 TO SN (°) | 108.2 ± 5.4 | 105.7 | 116.0 |
| L1 TO NB (mm) | 6.6 ± 2.8 | 6.5 | 3.5 |
| L1 TO MP (°) | 96.8 ± 6.4 | 82.0 | 72.8 |
| FACIAL ANALYSIS |  |  |  |
| E-LINE UL (mm) | −1.1 ± 2.2 | −3.5 | −1.5 |
| E-LINE LL (mm) | 0.5 ± 2.5 | 2.5 | 3.0 |

The patient reported that others in her family had prognathic mandibles. She first noted the anterior crossbite when her permanent incisors erupted at about age 6. Both genetic and environmental factors contributed to her Class III malocclusion.

### 3.2. Treatment Objectives

The treatment goals included correction of the anterior crossbites and establishment of a Class I dental relationship. It was also important to harmonize her facial profile by increasing the upper lip support and retracting the lower lip. Restoration of secondary caries on the restorations and a missing tooth were planned along with the orthodontic treatment.

### 3.3. Treatment Alternatives

Three options were proposed.

Option 1: Extraction of the maxillary first premolars, two-jaw orthognathic surgery to advance the maxilla, and setback the mandible with an optional genioplasty.

A combination of orthognathic surgery and orthodontic treatment may provide the best possible treatment results. However, the patient, in this case, rejected the surgical approach.

Option 2: Extraction of the maxillary second premolars, mandibular first premolars, and mandibular third molars.

The treatment goal for maxillary second premolar extraction would be to relieve maxillary anterior crowding. The purpose of the mandibular first premolar extraction would be to correct the anterior crossbites. The proposed extraction pattern was made considering the anchorage for molar Class III correction.

Option 3: Mandibular third molar extractions and full mandibular arch retraction to correct the anterior crossbites.

TSADs would be used to retract the whole mandibular dentition. After correcting the anterior crossbites, TSADs would be used to simultaneously retract the maxillary and mandibular dentitions.

After thorough discussion and communication, treatment plan Option 3 was accepted, understanding that a reevaluation would be made after the occlusion was corrected. If the profile were too protrusive, four premolar extractions would be considered the alternative treatment plan.

Regarding the missing maxillary right second molar, an implant prosthesis was proposed. The patient could decide on the orthodontic treatment.

### 3.4. Treatment Progress

After the removal of the mandibular third molars, orthodontic treatment commenced.

The maxillary arch was bonded with Damon 2 brackets, and a 0.014-in Cu-NiTi archwire was inserted as the initial archwire. A custom-made bite turbo was bonded on

the lingual surface of the mandibular left central incisors to prevent interference between the upper brackets and the mandibular dentition. Two months later, the mandibular arch was bonded with Damon 2 brackets, and a 0.014-in Cu-NiTi archwire was inserted. Two TSADs (A1-J, Bio-Ray Biotech Corporation, Taipei, Taiwan), 2.0 mm in diameter and 12 mm in length, were installed at the buccal shelves of the mandible on both sides. Two 150 g NiTi coil springs were attached from the head of the miniscrews to the brackets on the mandibular canines (Figure 11). Maxillary and mandibular archwires were changed to $0.016 \times 0.025$-in Cu-NiTi in the third month. One month later, the anterior crossbite was corrected to an edge-to-edge relationship. Elastomeric chains from mandibular canine to canine, combined with coil springs from the buccal shelf, TSADs to mandibular canines, were used to retract the whole mandibular dentition. Provisional crowns were fabricated on the four maxillary incisors to repair the severe incisal wear from the previous malocclusion. Class I dental relationships were achieved after four months of retracting the mandibular dentition with the bilateral buccal shelf TSADs. The patient complained of lip protrusion after anterior crossbite correction, so IZC (infrazygomatic crest) miniscrews were installed on both sides to distalize the whole maxillary dentition (Figure 12). Two months later, the archwires were sectioned, and the posterior segments of the archwires were removed. Up-and-down elastics (Ostrich, Ormco Co., Orange, CA, USA) were prescribed to optimize the interdigitations. After settling, all appliances were removed. The treatment duration was 12 months. The patient was satisfied with her treatment results, as documented in Figure 13. Maxillary and mandibular clear retainers were delivered, and the patient was instructed to wear the retainers full-time for the first 6 months and just at night thereafter. The retainers were renewed after the permanent prostheses were fabricated. Eight-year, three-month posttreatment records showed satisfactory stability (Figure 14).

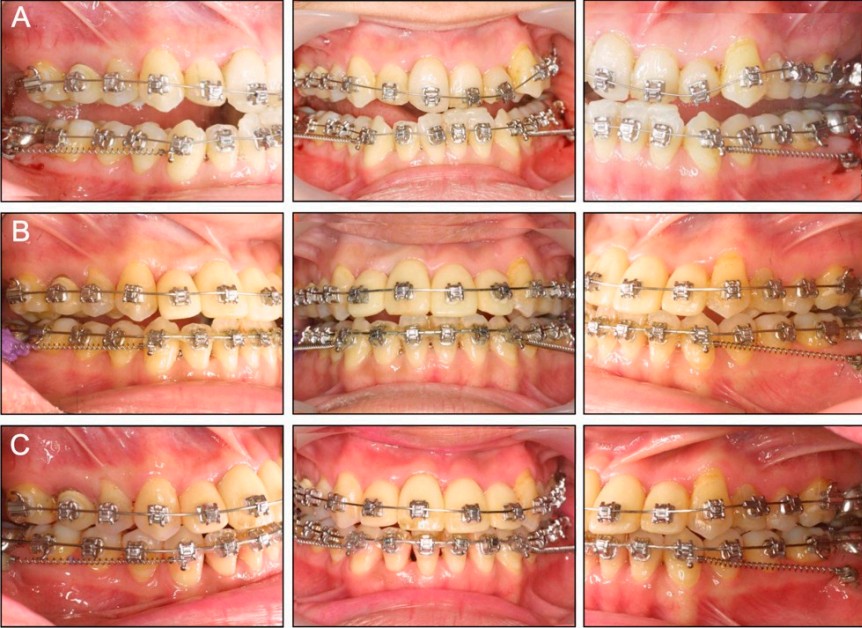

**Figure 11.** Case 2, Progressive records at (**A**) 2nd month (**B**) 4th month (**C**) 6th month show the correction of anterior crossbite and Class III malocclusion.

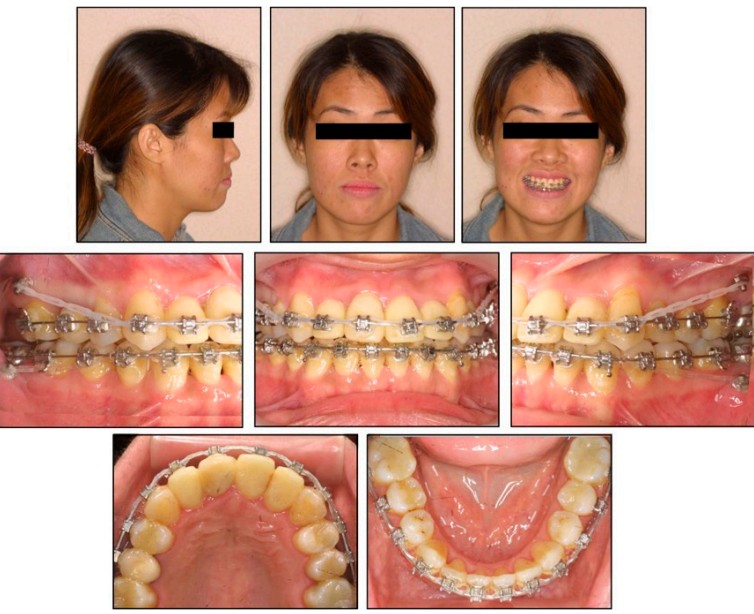

**Figure 12.** Case 2, Two upper posterior miniscrews were installed in the eighth month for the protrusion after the correction of anterior crossbite.

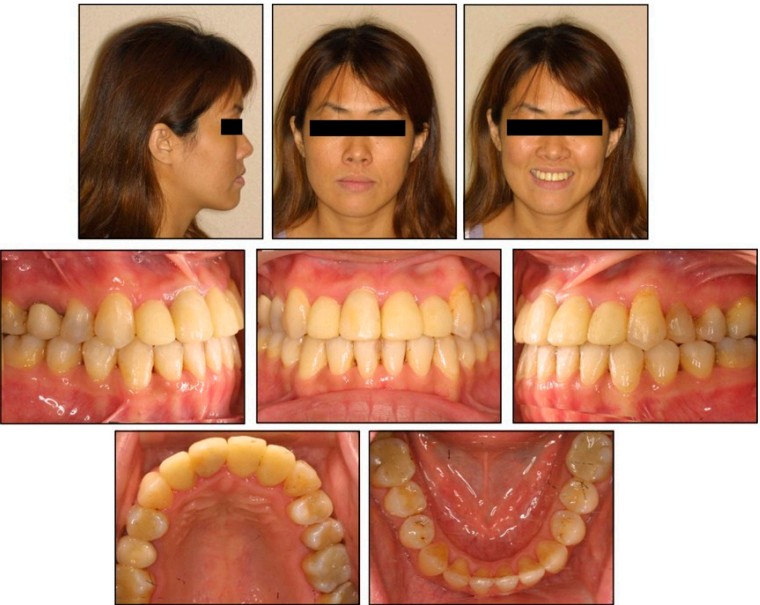

**Figure 13.** Case 2, Posttreatment facial and intraoral photographs.

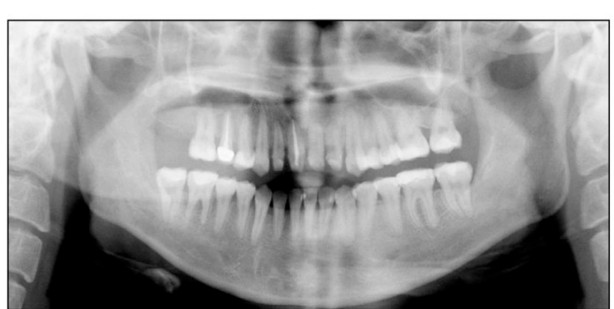

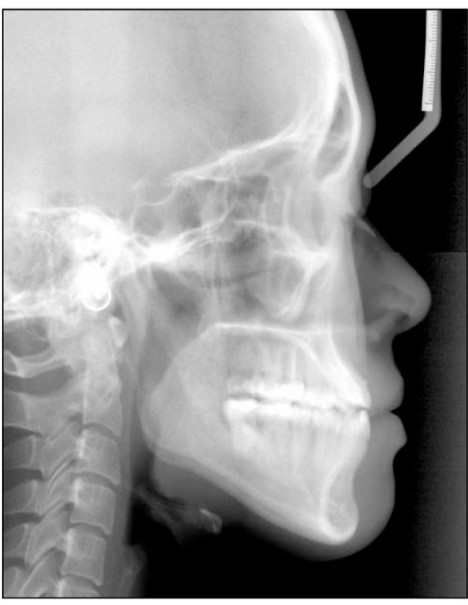

**Figure 14.** Posttreatment panoramic radiograph and lateral cephalogram.

*3.5. Treatment Results*

The anterior crossbites and Class III dental relationships were corrected, and the facial profile was improved (Figure 13). The posttreatment panoramic radiograph revealed acceptable root parallelism and good maintenance of the supporting alveolar bone (Figure 14).

The posttreatment lateral cephalogram indicated an acceptable orthognathic profile (ANB, −3°; Wits, −11.4 mm). The skeletal response was typical for camouflage treatment of skeletal Class III malocclusion, including an increased vertical dimension, proclination of the maxillary incisors (U1 -SN: from 105.7° to 116.0°), and retroclination of the mandibular incisors (IMPA: from 82.0° to 72.8°). A lateral cephalometric analysis showed that the mandibular plane angle was increased by 1.3° (MPA: from 28.5° to 29.8°). Clockwise mandibular rotation was noted in the cephalometric superimpositions (Figure 15; Table 2). The Follow-up records after 8 years and 3 months showed fairly good stability of the overall occlusion (Figure 16).

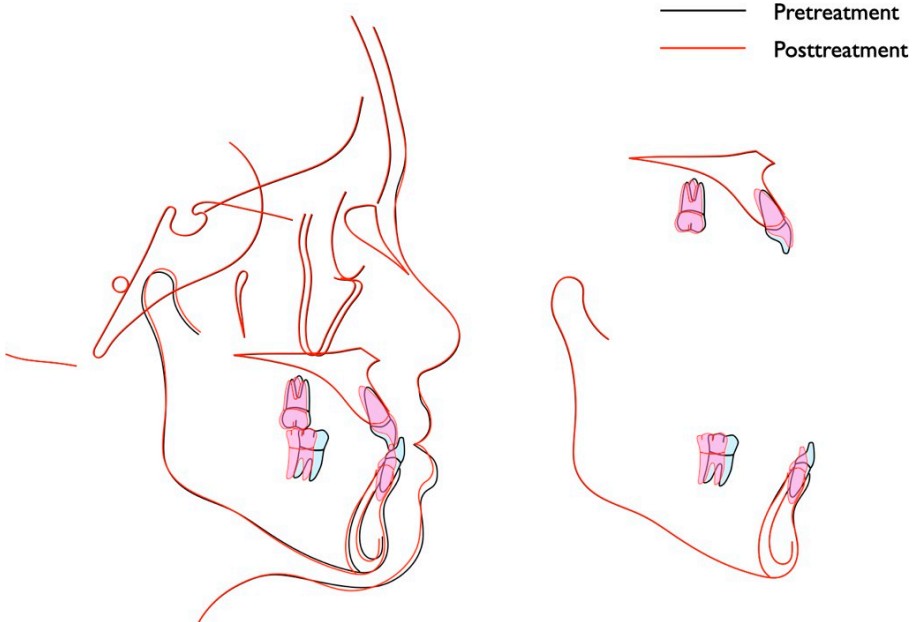

**Figure 15.** Case 2, Cephalometric superimpositions show the correction of anterior crossbite and Class III malocclusion with a nonextraction approach supported with TSADs.

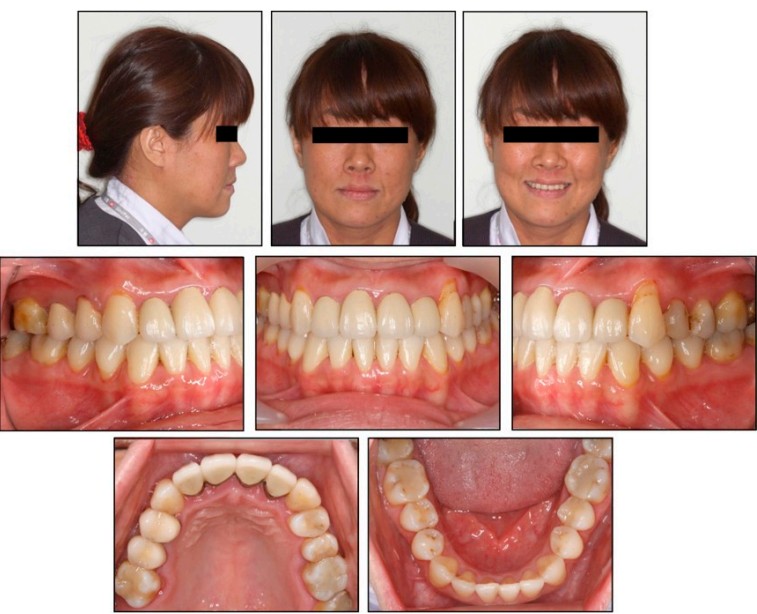

**Figure 16.** Case 2, Follow-up records after 8 years and 3 months.

## 4. Discussion

Dentoalveolar protrusion consequent to anterior crossbite correction is quite often in borderline Class III cases. The major tooth movements of anterior crossbite correction are advancement and proclination of maxillary incisors, while the retraction of the mandibular incisors is relatively limited. The choice between extraction and non-extraction may lead to a significant difference in the posttreatment profile [48]. An extraction treatment plan may avoid posttreatment protrusion, but risks with a dished-in posttreatment facial profile, especially in an adolescent patient with growth potential. Although the CVM (cervical vertebral maturation) of Case 1 was stage 5, indicated at least one year after the growth peak, late mandibular growth was possible in Case 1 to worsen the facial profile [49–52]. If orthognathic surgery is considered later after growth completion, there would be a significant round trip for dental decompensations before surgery. Hence, the nonextraction approach, in the beginning, might be more conservative, such as in Case 1. After correcting the anterior crossbite, a reevaluation can be made to check the patient's perception of the facial profile change. Four-premolar extraction or whole arch distalization might be considered if protrusion is a concern.

With four-premolar extraction, more retraction can be expected. It will take longer to close the extraction spaces. If the patient hesitates to extract premolars, extra-radicular miniscrews at IZC and buccal shelf for whole arch distalization might be another option, as in Case 2.

The overall cephalometric superimposition is barely able to show the true value of this treatment modality; the serial cephalometric superimpositions and profile photographs need to be examined, too (Figures 17 and 18). In the camouflage treatment of a Class III malocclusion, the mandibular arch is usually the limiting factor in the treatment result.

It is more difficult for mandibular dentition distalization than maxillary dentition advancement to correct a negative overjet, so it is common to see proclined maxillary incisors along with retroclined mandibular incisors after Class III camouflage treatment. Furthermore, the posttreatment profile usually looks protrusive. If the whole mandibular dentition can be distalized with miniscrews during treatment to avoid proclination of the maxillary incisors, the posttreatment profile might not look so protrusive.

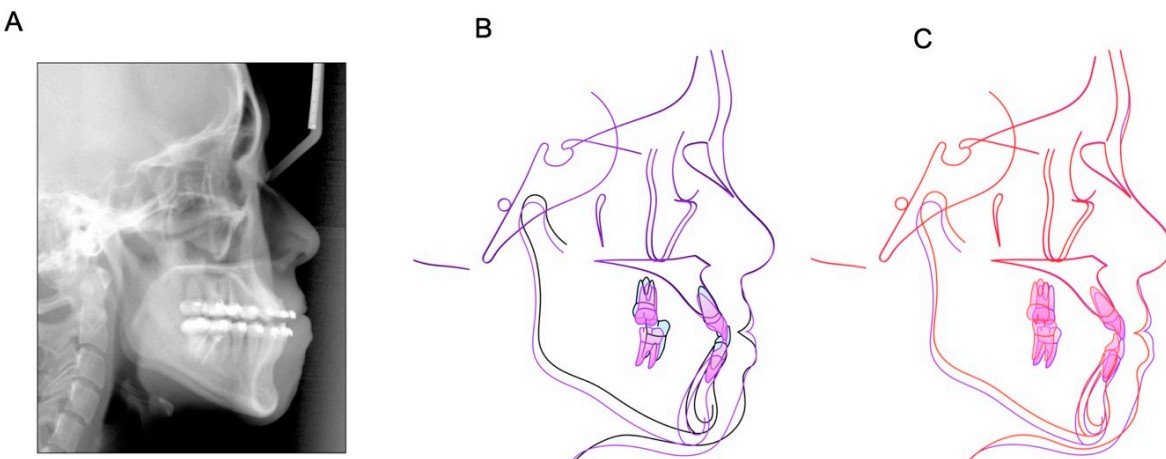

**Figure 17.** Case 2, (**A**), Lateral cephalogram in the 4th month of treatment right after the correction of anterior crossbite; (**B**), Cephalometric superimpositions of pretreatment and the 4th month show that most of the correction of anterior crossbite was due to proclination of the upper incisors, even though the mandibular arch was distalized a bit with TSADs on the buccal shelves; (**C**), Cephalometric superimpositions of posttreatment and the 4th month shows full arch distalization of both maxillary and mandibular arches to reduce the protrusion after anterior crossbite correction.

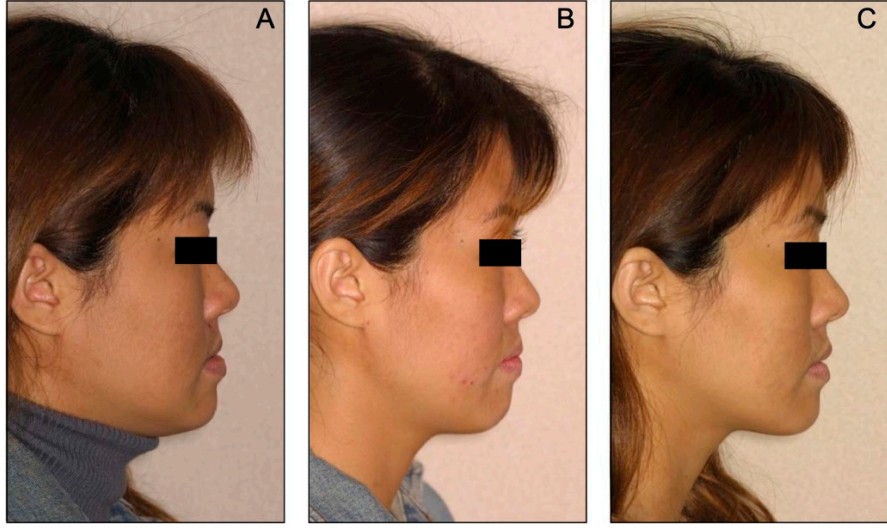

**Figure 18.** Case 2, profile comparisons. (**A**), Pretreatment; (**B**), Nonextraction approach with the mandibular arch distalization with TSADs; (**C**), Posttreatment with full arch distalization with TSADs on both arches.

In Case 2, we can see that even with buccal shelf TSADs, the amount of maxillary incisors advancement is more than the amount of distalization of the mandibular incisors because of the nature of the tooth movement. Labial tipping of the maxillary anterior teeth is much easier than retracting the whole mandibular dentition. That is why the profile looked protrusive after anterior crossbite correction. In this case, we installed another two TSADs at the bilateral infrazygomatic crests to distalize the whole maxillary dentition. Together with the two TSADs at the buccal shelves to distalize the whole mandibular dentition, a more orthognathic posttreatment profile was obtained. We reduced the treatment time to 12 months in Case 2 with the nonextraction approach and TSADs. Nonextraction treatment of some Class III malocclusions with TSADs is conservative and efficient.

The criteria for case selection in this approach would be the same as those with camouflage treatment of Class III malocclusions, including (1) mild to moderate skeletal discrepancy, (2) none or minor dental compensation, (3) an acceptable profile except for perioral imbalance,

and (4) sufficient clearance for whole dentition distalization. The third molars should be removed before whole arch distalization. The skeletal boundaries of the dentition should be checked carefully. Sugawara et al. reported the average amount of distalization of the maxillary first molar to be 3.8 mm at the crown level and 3.2 mm at the root level and also reported that the average amount of distalization of the mandibular first molar to be 3.5 mm at the crown level and 1.8 mm at the root level [33,34]. Kim et al. suggested that 3 mm of mandibular arch distalization can be expected in Class III malocclusion cases and highlighted the mandibular posterior anatomic limit (MPAL) to be the lingual cortex of the mandibular body [53]. Examination with cone-beam computed tomography (CBCT) might be needed for precise prediction. Aside from the mandibular posterior limit, the maxillary and mandibular lingual cortical plates and maxillary tuberosity determine the potential boundaries of tooth movement. Soft tissue boundaries of dentition should also be observed clinically. However, the mandibular posterior anatomic limit might be the most critical limiting factor with this treatment modality. In addition, patients need to be informed that the camouflage Class III treatment is primarily dental compensation of skeletal discrepancies, which might not be as ideal as the treatment results with the surgical approach.

The positions of TSADs at the buccal shelves are extra-radicular rather than interradicular (Figure 19) [38,43,47]. This is critical for this nonextraction approach, so the TSADs do not interfere with the root movement. The implant sites on the mandible are at the buccal shelves between the mandibular first and second molars. The flatter platform of the buccal shelf and more attached gingiva make this location favorable for placing a miniscrew. We suggest limiting the insertion point to somewhere within the mucogingival line. Suppose the attached gingiva is not wide enough for favorable miniscrew insertion. In that case, it is advised to insert the miniscrew as close to the mucogingival line as possible (within 1 mm) to minimize possible soft tissue irritation. The TSADs should be inserted perpendicular to the platform as possible on the buccal shelf, parallel to the mandibular molar roots, and into the bone until the proper amount of head exposure is achieved. The area only requires local infiltration of analgesia without any flap or pilot drilling, so a screwdriver is the only tool that will be necessary. Stainless steel miniscrews are preferred for safety reasons to avoid breakage during self-drilling. For more difficult insertions, a high-speed diamond round bur might be needed to make an indentation to catch the tip of the miniscrew for buccal shelf TSADs insertion. The insertion technique is quite simple and safe.

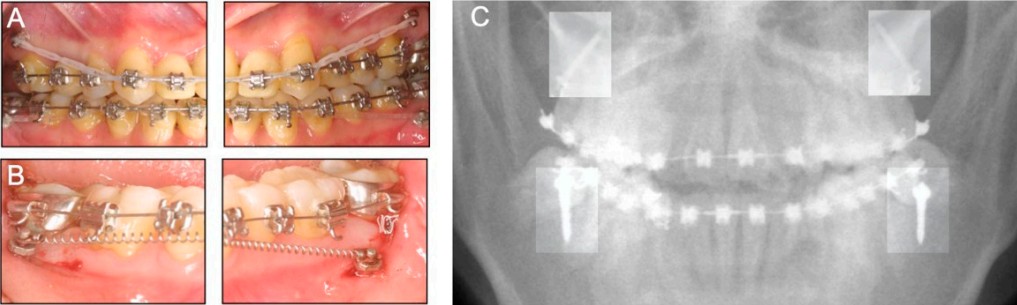

**Figure 19.** Extra-radicular positions of the TSADs at (**A**) infra-zygomatic crests and (**B**) buccal shelves are important for full arch distalization. The PA cephalogram (**C**) clearly shows the positions of the TSADs outside the dental roots.

The insertion points for the infrazygomatic crest TSADs are the attached gingival area of the maxillary molars ranging from the mesial to distal interdental areas of the maxillary first molars. In this case, the miniscrews were initially inserted perpendicular to the bony surface. After initial engagement with the cortical bone, the miniscrews were redirected to about 60° to the occlusal plane to avoid the roots and were aimed at the infrazygomatic crest. The selection of insertion points depends on the individual anatomy relative to bone quantity and quality [46].

The bite turbo used at the initial stage of this treatment not only prevented occlusal interference with the upper brackets but also helped with the intrusion of the mandibular incisors and extrusion of the maxillary posterior teeth, which subsequently rotated the mandible slightly backward [54].

## 5. Conclusions

The nonextraction approach for correcting anterior crossbites in the camouflage treatment of Class III malocclusions often leads to a protrusive profile and flared maxillary incisors. With the help of TSADs in Class III correction, the entire mandibular dentition can be distalized along with the maxillary dentition to avoid a protrusive profile after the correction of anterior crossbites. It is not only effective but also efficient with appropriate differential diagnosis and case selection. The extra-radicular positions of the miniscrews, instead of inter-radicular positions, are important for the success of this in whole arch distalization treatment approach. The boundaries of orthodontic tooth movement need to be carefully evaluated to avoid violation.

**Author Contributions:** J.J.-L.L. contributed to treating patients and writing. J.H.P. contributed to the reviewing and writing the case. All authors have read and agreed to the published version of the manuscript.

**Funding:** This research received no external funding.

**Institutional Review Board Statement:** The case report did not require ethical approval.

**Informed Consent Statement:** Written informed consent has been obtained from the patient(s) to publish this paper.

**Conflicts of Interest:** The authors declare no conflict of interest.

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
