# Peer review of "Nonsurgical Correction of Anterior Crossbite with Extra-Radicular Miniscrews—A Case Report"

_applsci, doi:10.3390/app122211719_

Round 1
Reviewer 1 Report
This is a very interesting article.
I would suggest the authors to cover the eyes of the patients in order to mentein their identity hidden.
In the discussion chapter the authors have to include more recent published articles.
I suggest you some articles
DOI
10.3390/app12073259
DOI
10.3390/app12052657
DOI
10.3390/medicina58050603
DOI
10.3390/medicina57121350
DOI: 10.23812/21-2supp1-27
Author Response
I would suggest the authors to cover the eyes of the patients in order to mentein their identity hidden.
~ The eyes were covered in the revised manuscript.
In the discussion chapter the authors have to include more recent published articles.
~ Thank you for the suggestions. More recent references were added in the manuscript.
Reviewer 2 Report
Dear Authors,
The aim of the present study was to illustrate two similar Class III anterior crossbite cases, treated with extraction and miniscrews, in order to show that the second type of approach could lead clinician to avoid extractions.
I would like to congratulate with authors for the conducted treatment.
The study is of scientific interest and in line with the aims of the journal.
However, there are some issues that should be added.
Abstract
Please follow the journal guideline (https://www.mdpi.com/journal/applsci/instructions).
The abstract should be a total of about 200 words maximum. The abstract should be a single paragraph and should follow the style of structured abstracts, but without headings: 1) Background: Place the question addressed in a broad context and highlight the purpose of the study; 2) Methods: Describe briefly the main methods or treatments applied. Include any relevant preregistration numbers, and species and strains of any animals used. 3) Results: Summarize the article's main findings; and 4) Conclusion: Indicate the main conclusions or interpretations. The abstract should be an objective representation of the article: it must not contain results, which are not presented and substantiated in the main text and should not exaggerate the main conclusions.
Introduction
I suggest improving Introduction section that is too short. Report definition of class III, etiology, epidemiology, common treatments (surgical and compensation tretaments).
Discussion
In both cases, supposing you did not have the hand and wrist rx, I suggest to clarify the stage of skeletal maturation according to CMV (Baccetti), which is known to be a reliable method compared to hand wrist. Please cite (Ferrillo et al. Reliability of cervical vertebral maturation compared to hand-wrist for skeletal maturation assessment in growing subjects: A systematic review. J Back Musculoskelet Rehabil. 2021;34(6):925-936. doi: 10.3233/BMR-210003.).
This because the two case reports were different according to skeletal maturation. A relapse in the first case could occur. Please discuss this point.
References
Please follow the journal guideline (https://www.mdpi.com/journal/applsci/instructions).
Journal Articles:
1. Author 1, A.B.; Author 2, C.D. Title of the article. Abbreviated Journal Name Year, Volume, page range.
Reviewer 3 Report
Dear Authors,
This manuscript is interesting and fits the objectives of the journal; but it is necessary to review some points in order to improve the quality of the paper:
-Please be sure to use only keywords accordingly to medical subject headings (Mesh word) for a better indexing.
-I ask you to check the plagiarism of your article using specific sites to get a similitary report
-About the Title of the article, I suggest you to modify it and add the type of article
-First of all please add more background data in introduction section, furthermore, at the end of this section you should better state the main aim of the study. Maybe You could add a subparagraph called "Aim" or "objectives".
- It is necessary to revise the English used, because in several places the construction of the sentences is confusing.
- The introduction section is very short and is needed to add other references to increase the quality of the manuscript.
- Add recent references about the topic of the article, dwelling in the introduction on articles published in 2022; Preferably a published articles should be with 50 or more references
I suggest you some articles about Orthognatic expander, syndromic conditions that could be needed of interventions, and about orthodontics movement that will help you improve your article.
Dento-Skeletal Class III Treatment with Mixed Anchored Palatal Expander: A Systematic Review DOI: 10.3390/app12094646
Oral-facial-digital syndrome (OFD): 31-year follow-up management and monitoring PMID: 29460530
Application of vibrational spectroscopies in the qualitative analysis of gingival crevicular fluid and periodontal ligament during orthodontic tooth movement DOI 10.3390/jcm10071405
Please expand conclusion section with main results and future perspectives of this study
Thank You,
Kind Regards
Reviewer 4 Report
the work is just a case series with no scientific contribution to the international community. I do not consider it valid for its publication
Author Response
~Thank you very much for your comments. Although there might be no scientific contribution, the aim of this case report was to illustrate the value of TSADs on the profile esthetics after anterior crossbite correction. Even with the use of BS miniscrews, most of the correction of anterior crossbite was achieved by the flaring of maxillary incisors. To obtain an esthetic facial profile, IZC miniscrews are very important to reduce the protrusion subsequent to anterior crossbite correction. We believe this is a good example to apply the science in clinical orthodontics to optimize the treatment for patients’ benefit.
Reviewer 5 Report
Overall I enjoyed reading this nice case report. Congratulations to the Authors. Please find my a few minor comments as notes in the attached pdf.

Author Response
~Thank you for the kind comments. The manuscript was amended accordingly.
The occlusion looked almost like a solid Class I dental relationship after 8 months of treatment (Fig 3).
The patient complained of lip protrusion after anterior crossbite correction, so IZC (infrazygomatic crest) miniscrews were installed on both sides to distalize the whole maxillary dentition (Fig 12).
Round 2
Reviewer 2 Report
Authors modified the text accordingly to suggestions.
Please cite this recent systematic review when you cite the CVM maturation
Ferrillo M, Curci C, Roccuzzo A, Migliario M, Invernizzi M, de Sire A. Reliability of cervical vertebral maturation compared to hand-wrist for skeletal maturation assessment in growing subjects: A systematic review. J Back Musculoskelet Rehabil. 2021;34(6):925-936. doi: 10.3233/BMR-210003. PMID: 33998532.
Author Response
The cited references can be improved.
Please cite this recent systematic review when you cite the CVM maturation
~ A recent CVM systematic review has been added in the references. The numbers of the other references were adjusted accordingly.
“Although the CVM (cervical vertebral maturation) of case 1 was stage 5, indicated at least one year after the growth peak, late mandibular growth was possible in case 1 to worsen the facial profile.49-52“
“52. Ferrillo, M., Curci, C., Roccuzzo, A., Migliario, M., Invernizzi, M., & de Sire, A. Reliability of cervical vertebral maturation compared to hand-wrist for skeletal maturation assessment in growing subjects: A systematic review. J. Back Musculoskelet. Rehabil. 2021, 34, 925-936.”

Reviewer 4 Report
Low quality. My opinion did not change
Author Response
Moderate English changes required
~ The manuscript was revised again by a native English speaker.
Low quality. My opinion did not change
~ Thank you for your comments. I am sorry that this manuscript does not meet your expectations. Your points were well understood.